# Customisable Silicone Vessels and Tissue Phantoms for In Vitro Photoplethysmography Investigations into Cardiovascular Disease

**DOI:** 10.3390/s24051681

**Published:** 2024-03-05

**Authors:** Parmis Karimpour, Redjan Ferizoli, James M. May, Panicos A. Kyriacou

**Affiliations:** Research Centre for Biomedical Engineering, City, University of London, London EC1V 0HB, UK; parmis.karimpour@city.ac.uk (P.K.); redjan.ferizoli@city.ac.uk (R.F.); james.may.1@city.ac.uk (J.M.M.)

**Keywords:** CVD, PPG, arterial stiffness, cardiovascular disease, photoplethysmography, tissue phantom, vascular ageing

## Abstract

Age-related vessel deterioration leads to changes in the structure and function of the heart and blood vessels, notably stiffening of vessel walls, increasing the risk of developing cardiovascular disease (CVD), which accounts for 17.9 million global deaths annually. This study describes the fabrication of custom-made silicon vessels with varying mechanical properties (arterial stiffness). The primary objective of this study was to explore how changes in silicone formulations influenced vessel properties and their correlation with features extracted from signals obtained from photoplethysmography (PPG) reflectance sensors in an in vitro setting. Through alterations in the silicone formulations, it was found that it is possible to create elastomers exhibiting an elasticity range of 0.2 MPa to 1.22 MPa. It was observed that altering vessel elasticity significantly impacted PPG signal morphology, particularly reducing amplitude with increasing vessel stiffness (*p* < 0.001). A *p*-value of 5.176 × 10^−15^ and 1.831 × 10^−14^ was reported in the red and infrared signals, respectively. It has been concluded in this study that a femoral artery can be recreated using the silicone material, with the addition of a softener to achieve the required mechanical properties. This research lays the foundation for future studies to replicate healthy and unhealthy vascular systems. Additional pathologies can be introduced by carefully adjusting the elastomer materials or incorporating geometrical features consistent with various CVDs.

## 1. Introduction

In 2019, cardiovascular diseases (CVDs) accounted for 32% of global deaths, totalling 17.9 million lives lost [1]. Atherosclerosis, hypertension, myocardial infarction, and stroke are among the CVDs that become more prevalent with ageing, profoundly impacting the heart and vascular system [2,3,4,5]. Primarily, vascular ageing contributes to CVD, marked by alterations in the mechanical and structural features of the vascular wall. These changes lead to a decline in arterial elasticity and reduced arterial compliance [6,7,8]. The ageing process prompts the formation of collagen bridges, preventing elongation by replacing the elastin fibres, giving the vessel walls a less elastic structure, and resulting in arterial stiffness [9,10,11]. Therefore, the measurement and assessment of arterial stiffness becomes imperative in diagnosing vascular ageing.

Current non-invasive assessment techniques, such as pulse transit time (PTT), vascular ultrasound, and magnetic resonance imaging (MRI), are confined to clinical settings due to the expertise required for the imaging techniques or the need for multiple measurement points, as in the case of PTT [12,13]. More precise methods, such as angiography and computerised tomography angiography (CTA), require the injection of contrast agents [14]. While these procedures are accessible within clinical environments, referrals by a General Practitioner (GP) may introduce delays in cardiovascular health assessment. Therefore, the demand for more innovative technologies capable of the accurate and non-invasive assessment of vascular ageing is clear.

In vitro studies simulating vascular ageing in a controlled environment offer a promising path for developing new non-invasive technologies and signal processing algorithms. Previously, it has been discussed that PPG sensor technology has the potential to provide information that is directly related to vascular ageing [15]. Previously, researchers have created artificial vessels through techniques such as 3D bioprinting [16,17,18], microfluidics based on laminar flow [19,20], and coaxial scale-up printing [21]. Although these approaches have shown encouraging results, they have limitations. For example, they often rely heavily on specific types of inks for printing, and the technology involved in laminar flow-based microfluidics can be complex [22].

This paper describes the fabrication, mechanical testing, and PPG feature extraction of custom silicone vessels and tissue phantoms, utilising silicone additives to simulate the human vascular system in an in vitro setup. This builds upon previous methods outlined by The Research Centre for Biomedical Engineering, City, University of London. Previously, polydimethylsiloxane (PDMS) (Sylgard 184) (Dow Silicones, Barry, Wales, UK) was explored to create the initial vessels [23]. While this was successful, the formulations in PDMS silicone cannot be adjusted for customisable vessel elasticity. In this study, vessel fabrication and mechanical testing are performed using PlatSil Gel-10 (Polytek Development Corp., Easton, PA, USA) silicone, which shows more versatility in modifying vessel properties. This paper investigates the effectiveness of the PlatSil elastomer in vessel stiffness customisation to determine the precision of formulation adjustments and the range of elasticities possible, which is greater than previously reported [24]. Furthermore, this silicone is composed of a scatterer, which diffuses sensor light more similarly to human tissue. Furthermore, compared with the work published by Ferizoli et al. [25], this study explores mechanical tests on the fabricated vessels, including hardness, thickness, and elasticity measurements. The vascular properties for the full range of elastomer mixtures possible using the PlatSil silicone are presented. The fabricated custom vessels were embedded within silicone to mimic the surrounding tissue, creating vessel–tissue phantoms. The resulting phantoms were integrated into an in vitro model that emulated a cardiovascular system of the lower body, comprising a pulsatile pump and silicone tubing, for the measurement of PPG analysis and statistical analysis [26]. The setup was used to pump fluid around the system and observe differences in PPG signals corresponding to varying levels of arterial stiffness. This study aims to utilise the vessels produced and the in vitro setup constructed to investigate the utility of PPG sensors in capturing differences in arterial stiffness, as well as determine the precision and maximum range of elasticities attainable through silicone dip-coating.

## 2. Materials and Methods

Custom silicone vessels were created through a precise dip-coating procedure using an elastomer mixture, a process which is discussed in further detail below. Once fabricated, tensile testing was performed to evaluate the mechanical properties, while a digital microscope was used to examine vessel wall dimensions. The vessels were embedded into silicone tissue to form vessel–tissue phantoms. Compression tensile tests were then performed to assess tissue stiffness. The phantoms were then connected to a mechanical pulsatile pump for PPG recording and analysis.

### 2.1. Fabrication of Custom Vessels

To fabricate the custom vessels, PlatSil Gel-10 elastomer was used, and silicone was chosen for its durability. In contrast to latex, silicone maintains its form and can withstand high temperatures [27], a crucial quality for the curing stage of the dip-coating process. The mixture comprises two parts: Part A and Part B. Initially, Part A and Part B were individually weighed using a digital precision scale (Scout, Ohaus, Parsippany, NJ, USA). Once equal amounts were added, they were mixed thoroughly. To prevent premature curing, a retarder was added, extending the pour time. The addition of a 1% retarder of the total weight doubled the pour time [24]. Depending on the application, different softness and hardness of the elastomer solutions can be achieved by adding Smith’s Deadener (Polytek Development Corp., Easton, PA, USA) or hardener, respectively. The silicone was then mixed and placed in a vacuum chamber (Applied Vacuum Engineering, Bristol, UK) for three minutes to remove air bubbles, before being utilised in the dip-coating process.

### 2.2. Dip-Coating Process

To construct the vessels, the elastomer mixture was poured into a silicone pot. The dip-coating method was set up as illustrated in Figure 1 and followed the methodology created by Nomoni et al. [28]. The commercial silicone tubing, employed as the form (Hilltop Products Limited, Warrington, UK) for the custom vessels, was positioned on pulleys and threaded through a heating element by the Precision Dip Coater (Qualtech Products Industry, Denver, CO, USA). One end of the form was attached to a tension weight, while the other side was secured to the dip-coater arm. The heating element was activated for three minutes, reaching a temperature of 275 °C. The elastomer was then poured into the silicone pot, and a coating layer of the elastomer mixture was applied by pulling up the form at a fixed speed of 40 mm/min, enabling the tubing to pass through without premature curing. The heating element was deactivated once the arm had reached its highest point, and the dip-coated tubing was left to cure. The internal form was carefully separated from the cured custom vessel.

The formulation of the mixture in the dip pot was adjusted by varying the amounts of retarder to generate vessels with different wall thicknesses, using a form with a total outer diameter (OD) of 3 mm. Similarly, deadener and hardener were incorporated to produce vessels with varying elasticities, using a form with OD of 2.6 mm.

### 2.3. Vessel–Tissue Phantom Fabrication

The surrounding tissue was formulated by combing equal amounts of Part A and Part B of Gel-00 (Polytek Development Corp., Easton, PA, USA). Gel-00, softer than Gel-10, was selected to achieve tissues with a Shore hardness of OO 30, in line with the mechanical properties of the surrounding tissue [29]. The mixture was then placed in a vacuum chamber to eliminate air bubbles before being poured into 3D-printed moulds, resulting in the creation of vessel–tissue phantoms.

### 2.4. Measuring Thickness

To investigate the impact of the formulations on the vessel wall thickness, sections of the constructed vessels, each 1 mm in length, were sliced from both ends. These sections were then stained with black paint to enhance viewing contrast and examined under a digital microscope (Celestron, Torrance, CA, USA) for wall thickness measurements, as illustrated in Figure 2. The sample size comprised three stained cross-sections from each side of the vessel, totalling six cross-sections for each vessel. Still pictures were captured using Celestron MicroCapture Pro software version 2.5 (Celestron, Torrance, CA, USA). The wall thickness of each section was measured four times and averaged. The mean value for each vessel was determined by finding the average between both ends of the vessels. The same process was employed to determine the internal diameter (ID). The OD was calculated based on ID by adding the thickness on each side of the vessel wall.

### 2.5. Measuring Elasticity

The Universal Testing System (Instron 5944, Norwood, MA, USA) was used in a tensile test configuration to measure the Elastic Modulus of each custom vessel, as shown in Figure 3. This involved two grips holding a vessel with a length of 15 cm in place and vertically drawing the sample apart with a drawing speed of 500 mm/min until the material reached its breaking point. This procedure was set in accordance with the ASTM D412 standard [30]. Along with the previously calculated vessel diameter and thickness measurements, each vessel was universally loaded with a force of 0.10 N to calculate Young’s modulus using Bluehill Universal software version 4.42 (Instron, Norwood, MA, USA). This process was repeated for each vessel to observe differences in elastic properties with varying levels of retarder, deadener, and hardener. For the tissue samples, the system was set to the compression test configuration (as seen in Figure 3) to measure the Elastic Modulus of each tissue phantom. Each tissue phantom measured 50 × 20 × 20 mm (L × W × H). The phantoms were placed on a flat metal plate and compressed to calculate Young’s modulus. A drawing speed of 12 mm/min was set in accordance with the ASTM D575-91 standard [31].

### 2.6. Measuring Hardness

The AD-100-OO Precision Shore Durometer (Checkline Europe, Enschede, The Netherlands) was used to assess the hardness of the cured silicone tissue from the dip pot, as depicted in Figure 4. The durometer was placed on the surface of the tissue with the footpad at a slight angle and then gently rolled to a flat position for the most accurate reading. Readings were taken from multiple sites on the tissue and averaged. This process was repeated for tissues with varying levels of retarder, deadener, and hardener. Consequently, a record was produced to observe changes in tissue hardness depending on additives.

### 2.7. Obtaining PPG Signals from an In Vitro System

The custom vessel–tissue phantoms were connected to an in vitro system consisting of a pulsatile pump and silicone tubing containing blood-mimicking fluid (BMF) connected to custom PPG and pressure sensors (PendoTech, Princeton, NJ, USA). The PD-1100 Pulsatile Pump System (BDC Laboratories, Wheat Ridge, CO, USA) served as a pulsatile flow source, replicating the human heartbeat and generating a pulsatile pressure waveform. The vascular system was constructed with silicone tubing, creating a network that served two functions; firstly, it was necessary to step down the vessel size, as the outlet diameter was greater than the diameter of the vessels of the phantom. Secondly, to recreate the distinctive pulsatile pressure waveform, various bifurcations and resistances were needed in the network to more accurately reflect the bifurcations and systemic resistances that are thought to give rise to various morphological features.

The fluid circulating within the system was a simple BMF, created by blending methylene blue powder (Thermo Fisher Scientific, Waltham, MA, USA with deionised water. This mixture increased absorption in the red and infrared regions of light, improving the PPG signal quality [32]. The custom vessel–tissue was positioned above a PPG sensor, comprising red and infrared LEDs with peak wavelengths of 660 nm and 940 nm, respectively, and a photodiode (BPW34, Osram, Munich, Germany) with a peak sensitivity of 900 nm. For signal processing and acquisition, the sensor was connected to a ZenPPG device [33], followed by a data acquisition card interfaced with a LabVIEW virtual instrument (National Instruments, Austin, TX, USA).

## 3. Results

The custom vessels were mechanically tested in order to understand the impact of the additives on the thickness and elasticity of the custom vessels. The hardness and elasticity of the tissue samples were also tested. The vessel–tissue phantoms were placed into the in vitro setup, and PPG signals were acquired, which will be discussed in more detail below.

### 3.1. Thickness, Internal Diameter, and Outer Diameter Measurements

Measurements of the wall thickness of the vessels made with varying amounts of retarder are presented in Figure 5. The findings indicate that increasing the retarder concentration reduced the wall thickness until stabilising at 3.0% and 3.5%. The ID ranged from 2.55 mm to 2.78 mm, and the OD ranged from 3.20 mm to 3.77 mm.

Table 1 and Table 2 respectively outline the wall thickness, ID, and OD values obtained when varying the deadener and hardener ratios within the elastomer. The wall thickness remained between 0.50 mm and 0.58 mm. This outcome was anticipated, as the amounts of deadener and harder should not affect the thickness of the vessel; instead, they influence its softness and hardness [24]. For both datasets, the retarder concentrations were maintained at 1.5%. The OD ranged from 3.96 mm to 4.04 mm for the deadener dataset, whereas in the hardener dataset, the OD ranged from 3.96 mm to 4.09 mm. For both the deadener and hardener experiments, equal amounts of Part A and Part B were used with varying amounts of Part D (deadener) and Part H (hardener). Deadener experiments are represented with the ratio A:B:D, up to a limit of 0.6 D, and hardener with A:B:H, up to 1 H.

### 3.2. Elasticity Measurements

The Young’s modulus measurements recorded during the tensile test of the custom vessels made with varying amounts of retarder are presented in Figure 6. The resulting Young’s modulus ranged from 0.42 MPa to 0.61 MPa with retarder concentrations of 1.5% to 3.5%, respectively.

The Young’s modulus calculations on vessels with varying amounts of deadener and hardener are shown in Table 3 and Table 4, respectively. As the ratio of deadener increased, the resulting Young’s modulus values decreased from 0.52 MPa to 0.20 MPa, indicating an increase in elasticity. Conversely, as the ratio of hardener increased, the Young’s modulus increased from 0.42 MPa to 1.22 MPa, indicating a reduction in elasticity. Figure 7 displays these elasticity trends between the deadener and hardener and shows the Young’s modulus of a healthy femoral artery for the adventitia and radial strain [34].

In the case of tissue phantoms, an increase in the amount of deadener corresponded to a softening of the material, as demonstrated by the decrease in the Young’s modulus illustrated in Figure 8. With increasing hardener, the Young’s modulus of the tissue phantoms rose, indicating greater hardness of the material. A normal Young’s modulus for tissue found in the finger is shown in Figure 8, which ranges between 0.07 MPa and 0.2 MPa.

### 3.3. Hardness Measurements

The hardness measurements obtained from the cured tissues with varying retarder levels using the Shore durometer are recorded in Table 5. Hardness levels remained constant as the retarder was increased from 1.5% to 3.5%, remaining between OO 47 and OO 50 Shore hardness.

With increasing amounts of deadener, a reduction in tissue hardness was observed, as indicated in Table 6. Shore hardness decreased from Shore OO48 with no deadener to Shore OO25 at one-part deadener. On the other hand, tissue hardness, as expected, increased with hardener concentration, as shown in Table 7. At one-part hardener, the tissue stiffness increased to Shore OO67. The change in Shore hardness with hardener and deadener and the Shore hardness found in the dorsal thigh are presented in Figure 9.

### 3.4. Acquisition of Photoplethysmography Signals

PPG signals were obtained from three custom vessel–tissue phantoms created with varying additives to manipulate vessel elasticity. Deadener was used as the additive in the elastic vessel at a ratio of 0.2 (10% of the total weight). For the stiff vessel, the same amount of hardener was used. PPG signals were also recorded from a phantom with a vessel containing no additive. The PPG and pressure signals obtained from the phantoms are shown in Figure 10. The PPGs visually resemble human signals, featuring diastolic and systolic peaks with a dicrotic notch. There was a visual change in the PPG signals among the phantoms. As vessel stiffness increased, the amplitude of the systolic and diastolic peaks appeared to decrease, particularly in the infrared signal.

### 3.5. Statistical Analysis

Systolic PPG amplitude was quantified through feature extraction and presented in a box plot in Figure 11, illustrating the amplitude change among the vessel–tissue phantoms. This change was analysed for statistical significance using Kruskal–Wallis analysis, as shown in Table 8. *p*-values of 5.176 × 10^−15^ for red and 1.831 × 10^−14^ for infrared indicate a significant difference in amplitude between the phantoms (*p* < 0.001). A Dunn test was applied as a post hoc test to determine the degree of difference between each phantom, as seen in Table 9. The *p*-values, adjusted for multiple comparisons, suggest that the PPG amplitude of all three phantoms differs significantly (*p* < 0.001).

## 4. Discussion

The research presented here puts forward an innovative way to investigate the effect of arterial stiffness on PPGs utilising novel vascular tissue phantoms. The process outlined in this paper can be adapted to introduce pathologies by modifying elastomer properties to mimic both healthy and diseased vessels. The results demonstrate that the vessel wall thickness can be regulated using different retarder concentrations. Additionally, the softness and hardness of the vessels and surrounding tissue can be adjusted by incorporating a deadener or hardener. Preliminary PPG signal analysis confirmed that variations in vessel stiffness within phantoms lead to significant amplitude changes.

Increasing the retarder concentration reduced the thickness of the custom vessels. This change in thickness is linked to the cure time, which is determined by the retarder level. When the retarder concentration was increased, the silicone coating had more time to descend, resulting in a thinner dip-coated vessel before curing. It should be noted that the retarder level did have an impact on vessel elasticity, as explained by the change in thickness. It was found that the Young’s modulus was thickness-dependent; as the thickness decreased, the Young’s modulus increased [37].

Deadener was used to produce softer and more elastic vessels, simulating the properties of healthy human blood vessels. When the concentration increased to a maximum of 0.6 parts deadener, Young’s modulus decreased from 0.52 MPa to 0.20 MPa, indicating an increase in elasticity. As expected, the wall thickness remained between 0.53 mm and 0.58 mm, showing that the deadener primarily impacted the material’s stiffness. When creating stiffer materials, hardener was utilised to decrease elasticity, mimicking unhealthy vessels. The Young’s modulus obtained using hardener ranged from 0.52 MPa to 1.22 MPa with one-part hardener, representing reduced elasticity. Again, the wall thickness did not alter significantly with the hardener, remaining between 0.50 mm and 0.54 mm.

In the silicone tissue, the deadener was found to have the same effect—making the silicone softer, as indicated by a decrease in shore hardness. Adding hardener had the opposite effect, causing the tissue to become stiffer, as shown by the increase in shore hardness. Therefore, the Young’s modulus and tissue hardness are proportional, as also found by Sun et al. [38]—as the hardness increased, the Young’s modulus also increased. The dorsal thigh exhibits a shore hardness ranging from shore OO 25 to shore OO 31 [36]. To replicate this, equal amounts of Part A, Part B, and deadener can be incorporated. The tissue in a finger has an average elasticity of between 0.07 MPa and 0.2 MPa [35]. By using a ratio of 0.2 to 0.8 deadener to equal amounts of Part A and Part B, it is feasible to create tissue of a similar elasticity to that of the human finger.

Peripheral arterial disease (PAD) is a restriction in the arteries, which usually occurs in the lower extremities [39]. As vessels age, their longitudinal and circumferential elasticity gradually diminishes [40]. The results of this study were compared with those of the common femoral artery. According to Soneye et al. [41], the wall thickness of a healthy left common femoral artery is 0.55 ± 0.05 mm. The common femoral artery’s vessel diameter ranged from 3.9 mm to 8.9 mm, with an average value of 6.6 mm [42]. A healthy femoral artery has been found to have a Young’s modulus of 0.80 MPa, 0.79 MPa, and 0.82 MPa for the intimate, adventitia, and radial strain, respectively [34]. From the results obtained in this study, the femoral artery can be mimicked with a 1A:1B:0.6H mixture with 1.5% retarder. If commercial silicone tubing with an OD of 2.6 mm is utilised as a form, a wall thickness of 0.51 mm can be attained. The Young’s modulus of 0.82 MPa is within the range for radial strain, and the diameter would be 3.96 mm, which is also within the normal range. A clogged femoral artery, known as an atherosclerotic femoral artery, has a Young’s modulus of 3.6 MPa and 2.11 MPa in the adventitia and radial strain, respectively [34]. As such, to mimic an atherosclerotic femoral artery, a higher ratio of hardener must be used than 1A:1B:1H to obtain a less elastic artery with a higher Young’s modulus. In this case, other materials, such as PlatSil Gel-25 (Polytek Development Corp., Easton, PA, USA), should be investigated to analyse the feasibility of creating stiffer vessels.

PPG signals were recorded from three phantoms composed of vessels with varying elasticities to determine if the elasticity influenced the PPG waveform, particularly the amplitude. The softer vessel phantom was created using a ratio of 0.2 deadener (10% of total weight) and the stiffer vessel phantom was made with hardener at a ratio of 0.2 (10% of total weight). The third vessel phantom did not contain additives. Red and infrared PPG signals were recorded and analysed on a cycle-by-cycle basis, which showed that as vessel stiffness increased, PPG amplitude decreased. Kruskal–Wallis analysis followed by a Dunn test found this change to be statistically significant (*p* < 0.001). This confirms the hypothesis that PPG signals can indicate changes in arterial stiffness, particularly through amplitude changes. This trend is as expected considering the physiology of blood vessels and PPG. Stiffer vessels are less elastic and therefore are less able to expand when the blood pulse wave arrives at the PPG sensor. As PPG is a volumetric technique, and the reduced expansion results in a lower change in the volume of blood at the site, the amplitude of the signal is lower.

Accompanying pressure signals confirmed that the morphology of the PPG wave was induced by changes in pressure. As with the PPG signals, the pressure signals obtained conformed to the expected physiological morphology. Interestingly, the varying vessel stiffness revealed a difference in the PPG morphology such as amplitude, while the pressure waveform remained constant across phantoms. This may be because the pressure sensors were placed after at the end of the vessel phantom, while the PPG sensors were positioned in the middle, possibly providing more information about blood flow changes within the vessel. While pressure waveforms serve as a secondary measure to validate the results, exploring additional methods, such as Doppler ultrasound, could provide further confirmation of the changes in volume.

To extend the findings outlined in this study to clinical measurements, it is crucial to assess potential influences on the PPG waveform. One such factor is sensor contact. While a secure and strong contact has proven to yield high-quality signals, peripheral vasoconstriction may result in signals of lower quality [15,43]. In the in vitro setup, the phantoms and sensors were housed within a custom casing that prevented movement and ensured consistent contact force during readings. Another factor to consider is hydrostatic effects. The positioning of a limb in relation to the heart has an impact on arterial flow and venous return, influenced by the hydrostatic effect. The research indicated that lowering the hand below heart level results in decreased amplitudes of both alternating current (AC) and direct current (DC) components of PPG, which has been attributed to venous distension. In contrast, raising the arm above heart level leads to an increase in AC and DC PPG amplitudes [44]. Therefore, it is concluded that the position of the hand is crucial in in vivo studies due to hydrostatic effects. The in vitro setup was aligned horizontally and phantom placement was kept constant. As such, minimal hydrostatic impacts were observed.

When fabricating custom vessels, some limitations were identified. Due to the delicate nature of the vessels, a ratio of deadener up to 1A:1B:0.6D is possible. It was observed that beyond this ratio, the custom vessel would adhere to the internal commercial silicone tubing being used as the form, making removal challenging. Furthermore, a minimum retarder of 1.5% should be used when the dip coater’s speed is set to 40 mm/min to allow for sufficient time to pull the vessel to the maximum height. Future work can explore the impact of varying the dip coater’s speed. One hypothesis suggests that increasing the speed may reduce the requirement for as much retarder, as it allows the uncured silicone to reach the maximum height quicker, thus shortening the need for a long curing time. Moreover, if the dip coater’s speed is increased while using the same amount of retarder, it is anticipated that the custom vessel will reach its maximum height more rapidly, potentially providing more time for the uncured silicone to descend due to gravity, resulting in thinner custom vessels. Furthermore, additional research should be carried out with a higher ratio of hardener to mimic the stiffness of an atherosclerotic femoral artery. The current experimental procedure was carried out using a deionised water-based fluid, representative of the absorption spectra of the blood [45]. In vitro investigations utilising non-Newtonian fluids would be beneficial to simulate the behaviour of human blood as it is composed of Casson fluid flow [46]. Signal analysis can be expanded upon by exploring additional parameters during PPG feature extraction. Features such as half peak width, area under the curve, upslope, and downslope are beneficial, as these characteristics may be indicative of arterial stiffness, providing a deeper understanding of the dynamics and mechanical properties of the custom vessels.

## 5. Conclusions

With recent developments in PPG, assessment of vascular ageing and CVD would benefit from an in vitro setup that can simulate both healthy and diseased vascular systems. This paper describes the construction of customisable novel silicone vessels, produced through a dip-coating procedure, to be used in an in vitro vascular system. The desired vessel thickness and stiffness attributes can be obtained by adjusting the elastomer formulations. By deliberately altering the formulations to influence the vessel properties, pathologies can be introduced into the vessels and compared to healthy vessels through PPG analysis. This can be employed in conjunction with in vivo studies.

This study aimed to create custom vessels with varying levels of stiffness for use in an in vitro model. In this model, PPG signals were recorded and analysed to detect morphological changes induced by increased vessel stiffness. The waveform of the resulting PPG signals varied depending on the stiffness of the vessel within the phantom. Statistically significant changes in amplitude were detected, indicating the possibility of measuring arterial stiffness with PPG technology.

This paper provides a procedure for making custom vessels and illustrates the changes in the morphology of the PPG signal, particularly amplitude, induced by elasticity changes to simulate arterial stiffness. The vessel manufacturing process can be improved by altering the dip-coating stage to enable the production of longer custom vessels. Other materials, such as PlatSil Gel-25, can also be explored to extend the stiffness of the custom vessels. Further analysis of the PPG waveform is necessary, by extracting a range of features from phantoms utilising non-Newtonian fluids, to unravel the link between PPG and arterial stiffness.

## Figures and Tables

**Figure 1 sensors-24-01681-f001:**
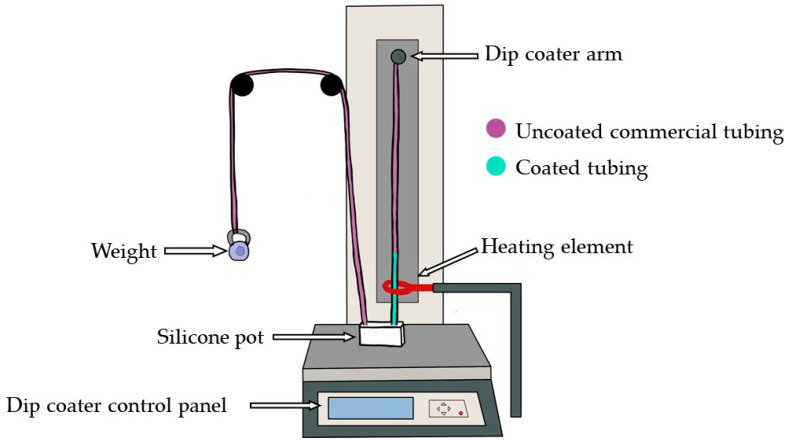
Schematic diagram of the dip-coating stage to produce customised vessels. The form is threaded through the silicone pot and heating element and attached to the dip coater arm. The dip coater arm pulls the form to apply a layer of elastomer from the silicone pot. Once cured, the form is separated, isolating the silicone coating as the custom vessel.

**Figure 2 sensors-24-01681-f002:**
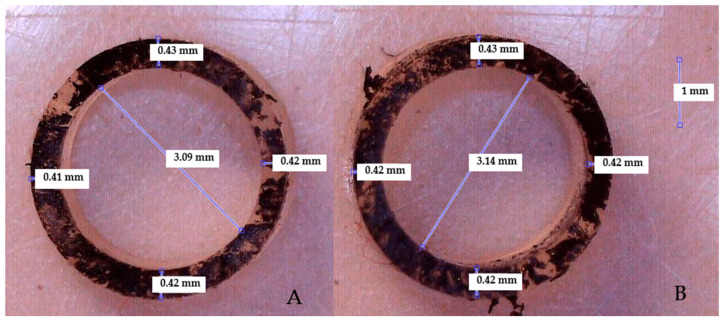
Cross-sectional image of a stained vessel during wall thickness measurement. (**A**) represents one end of the vessel and (**B**) represents the other end.

**Figure 3 sensors-24-01681-f003:**
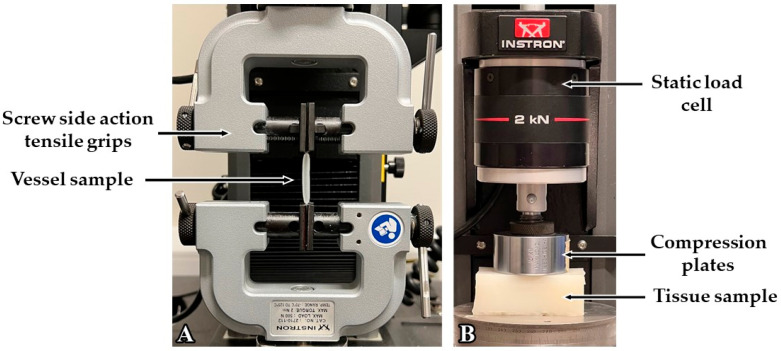
Universal Testing System for Elastic Modulus measurement of custom vessels through tensile testing (**A**) and compression testing of silicone tissue samples (**B**).

**Figure 4 sensors-24-01681-f004:**
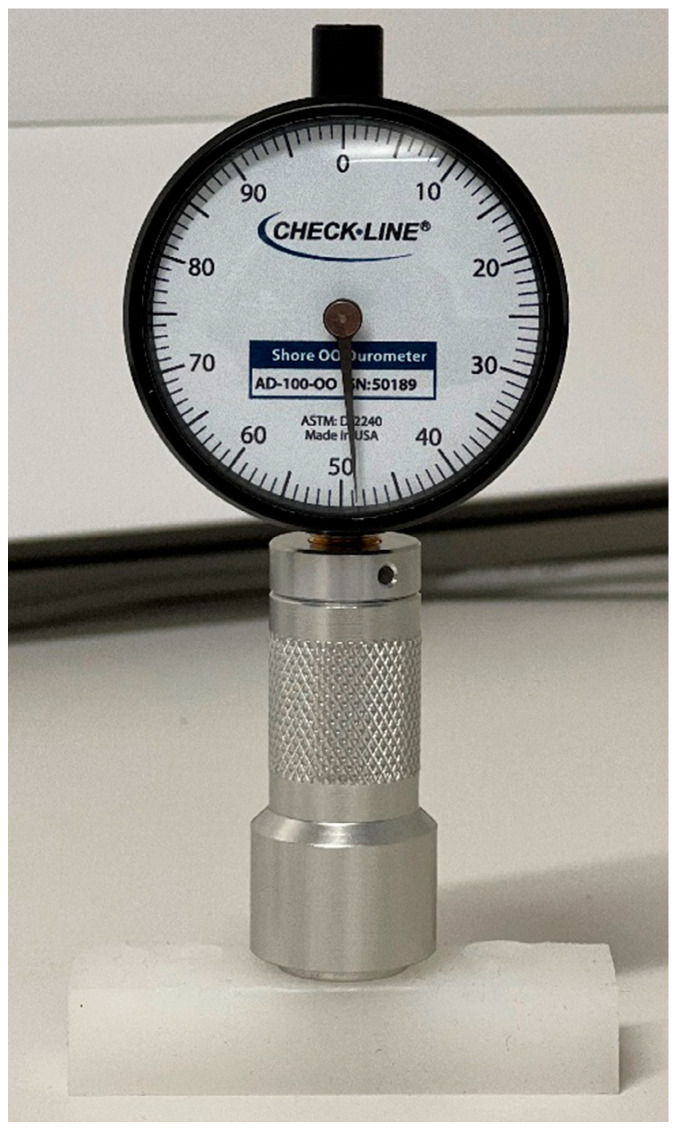
Measuring the hardness of silicone tissue using a Shore-OO Durometer.

**Figure 5 sensors-24-01681-f005:**
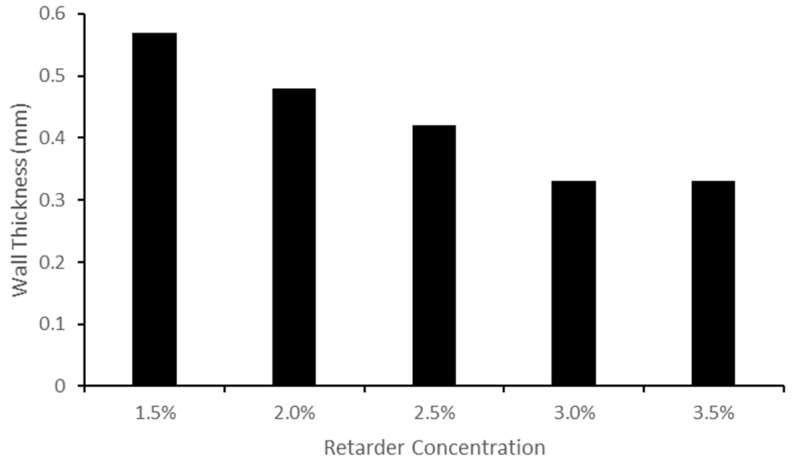
Bar chart displaying the change in wall thickness measurements for varying retarder concentrations.

**Figure 6 sensors-24-01681-f006:**
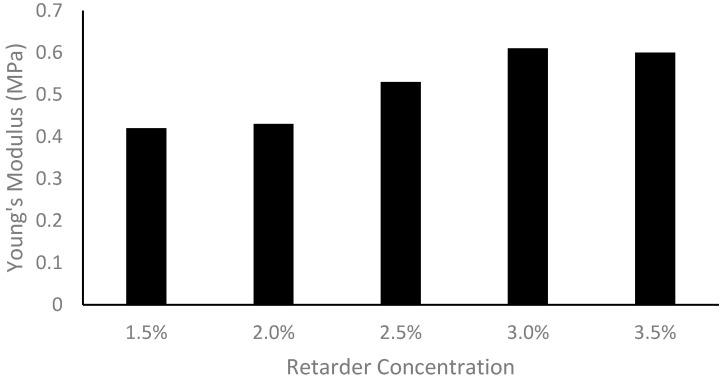
Bar chart displaying the Young’s modulus measurements for varying levels of retarder concentrations.

**Figure 7 sensors-24-01681-f007:**
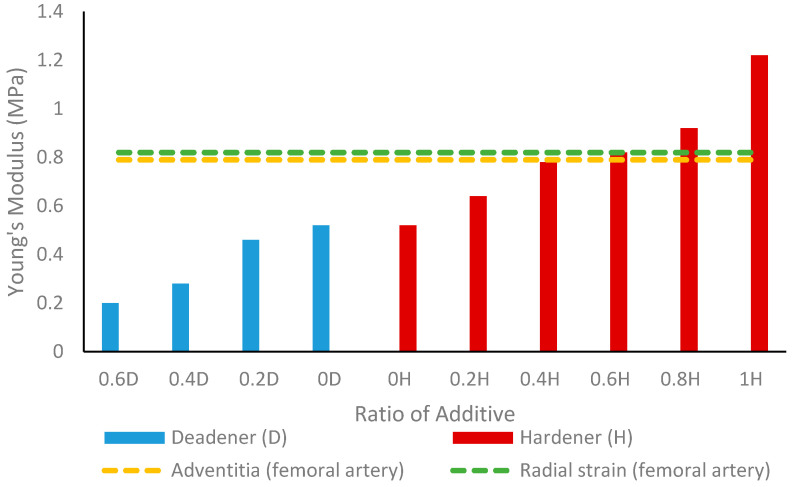
Tensile test results showing Young’s modulus values of custom vessels with different amounts of hardener and deadener. The ratio of additive is relative to the total weight of the elastomer while maintaining equal amounts of Part A and Part B. The Young’s modulus of a healthy femoral artery, in terms of the adventitia and radial strain [34], is shown using dotted lines.

**Figure 8 sensors-24-01681-f008:**
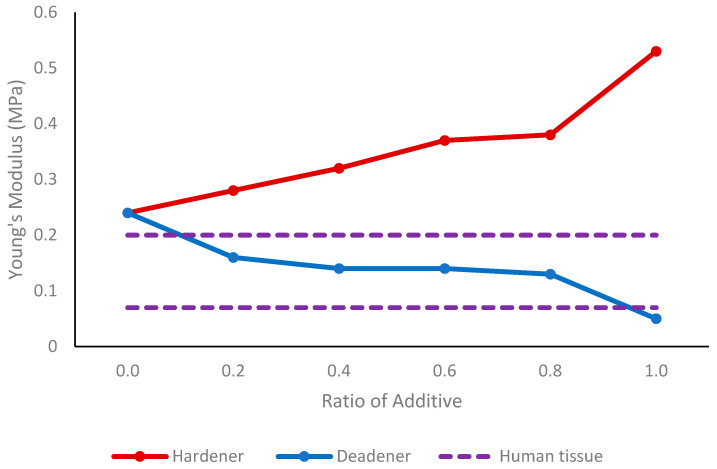
Compression test results showing Young’s modulus values of custom tissues with different amounts of hardener and deadener. The normal range for tissue found in the human finger [35] is indicated by the dotted lines.

**Figure 9 sensors-24-01681-f009:**
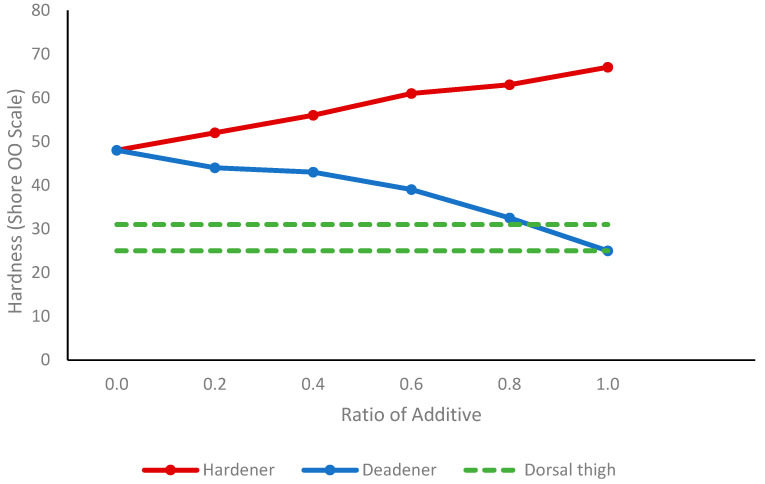
Change in Shore hardness with different amounts of hardener and deadener. The normal range for tissue found in the dorsal thigh [36] is indicated by the dotted lines.

**Figure 10 sensors-24-01681-f010:**
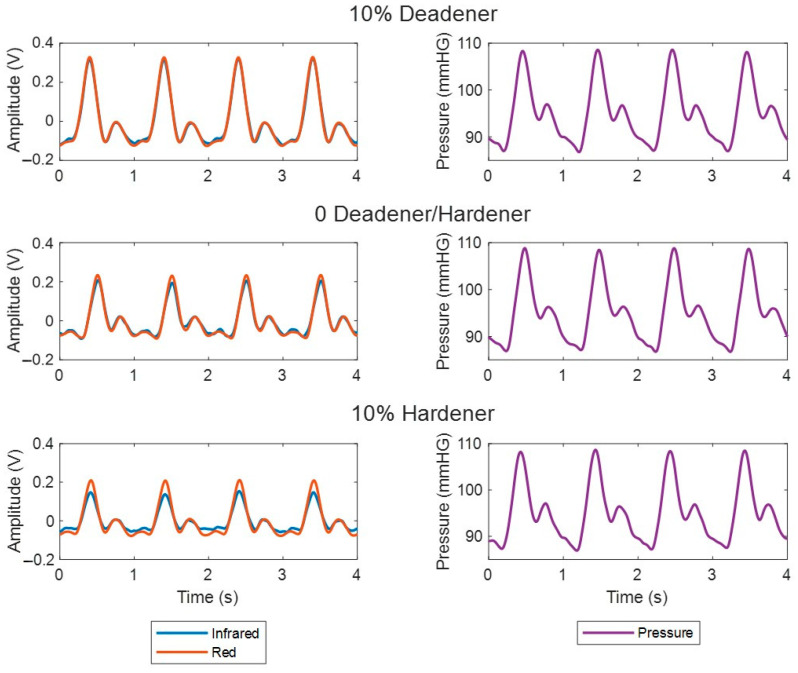
PPG and pressure signals recorded from custom vessel–tissue phantoms with vessels of varying stiffnesses induced by mixing different additives (hardener and deadener). The percentage of additive shown is relative to the total mixture.

**Figure 11 sensors-24-01681-f011:**
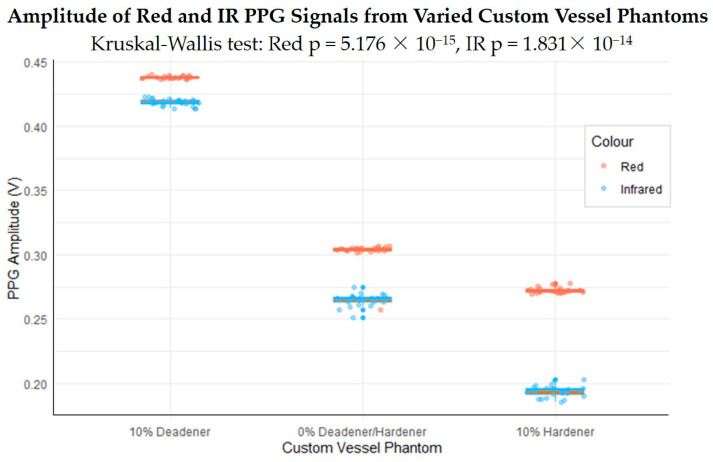
Box plot showing the change in amplitudes of red and infrared PPG signals from custom phantoms with vessels of varying elasticity.

**Table 1 sensors-24-01681-t001:** Thickness, internal diameter, and outer diameter measurements for varying deadener ratios. * Ratio of Part A:Part B:deadener.

Mixture Ratio (* A:B:D)	Wall Thickness (mm)	Internal Diameter (mm)	Outer Diameter (mm)
1:1:0	0.53	3.00	3.96
1:1:0.2	0.58	2.82	4.00
1:1:0.4	0.57	2.90	4.04
1:1:0.6	0.58	2.85	4.00

**Table 2 sensors-24-01681-t002:** Thickness, internal diameter, and outer diameter measurements for varying hardener ratios. * Ratio of Part A:Part B:hardener.

Mixture Ratio (* A:B:H)	Wall Thickness (mm)	Internal Diameter (mm)	Outer Diameter (mm)
1:1:0	0.53	3.00	3.96
1:1:0.2	0.54	2.98	4.06
1:1:0.4	0.50	3.02	4.02
1:1:0.6	0.51	2.95	3.96
1:1:0.8	0.53	2.96	4.01
1:1:1	0.52	3.02	4.09

**Table 3 sensors-24-01681-t003:** Young’s modulus measurements of custom-made silicone tubing with varying levels of deadener.

Mixture Ratio (A:B:D)	Young’s Modulus (MPa)
1:1:0	0.52
1:1:0.2	0.46
1:1:0.4	0.28
1:1:0.6	0.20

**Table 4 sensors-24-01681-t004:** Young’s modulus measurements of custom-made silicone tubing with varying levels of hardener.

Mixture Ratio (A:B:H)	Young’s Modulus (MPa)
1:1:0	0.52
1:1:0.2	0.64
1:1:0.4	0.78
1:1:0.6	0.82
1:1:0.8	0.92
1:1:1	1.22

**Table 5 sensors-24-01681-t005:** Young’s modulus measurements of custom-made silicone tubing with varying levels of retarder.

Retarder (%)	Hardness (Shore OO Scale)
1.5%	OO 48
2.0%	OO 49
2.5%	OO 49
3.0%	OO 50
3.5%	OO 47

**Table 6 sensors-24-01681-t006:** Shore hardness of tissue as the amount of deadener is increased, showing a negative correlation. As the ratio of deadener in the mixture is increased, the tissue hardness decreases due to the softening effect of the deadener.

Mixture Ratio (A:B:D)	Hardness (Shore OO Scale)
1:1:0	OO 48
1:1:0.2	OO 44
1:1:0.4	OO 43
1:1:0.6	OO 39
1:1:0.8	OO 32.5
1:1:1	OO 25

**Table 7 sensors-24-01681-t007:** Shore hardness of tissue as the amount of hardener is increased, showing a positive correlation. As the ratio of the hardener in the mixture is increased, the tissue hardness also increases.

Mixture Ratio (A:B:H)	Hardness (Shore OO Scale)
1:1:0	OO 48
1:1:0.2	OO 52
1:1:0.4	OO 56
1:1:0.6	OO 61
1:1:0.8	OO 63
1:1:1	OO 67

**Table 8 sensors-24-01681-t008:** Kruskal–Wallis analysis of red and infrared PPG amplitudes from custom phantoms.

Comparison	Chi-Squared	df	*p*-Value
Red	65.789	2	5.176 × 10^−15^
Infrared	63.263	2	1.831 × 10^−14^

**Table 9 sensors-24-01681-t009:** Dunn test of PPG amplitudes between each custom phantom.

Comparison	Z	*p* (Unadjusted)	*p* (Adjusted)
0 D/H–10% D	−4.055536	5.001957 × 10^−5^	1.000391 × 10^−4^
0 D/H–10% H	4.055536	5.001957 × 10^−5^	5.001957 × 10^−5^
10% D–10% H	4.055536	5.017550 × 10^−16^	1.505265 × 10^−15^

## Data Availability

Data are contained within the article.

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
