# Peer review of "Customisable Silicone Vessels and Tissue Phantoms for In Vitro Photoplethysmography Investigations into Cardiovascular Disease"

_sensors, 2024, doi:10.3390/s24051681_

Round 1

Reviewer 1 Report

Comments and Suggestions for Authors

This manuscript by Karimpour et al describes the fabrication of custom made silicon vessels with varying mechanical properties. The vessels were embedded within custom made tissue phantoms and were evaluated in an in vitro setting utilizing photoplethysmography reflectance sensors. This is an interesting study to explore how changes in silicone formulations influence vessel properties and their correlation with features extracted from photoplethysmography signals from vessels with different mechanical properties. Nevertheless, some issues should be settled before further consideration. Detailed comments were suggested as follows.

1. The abstract should state briefly the purpose of the research, the principal results and major conclusions. It should contain significant and quantitative findings.

2. It seems that the methodology of fabricating custom silicone vessels and tissue phantoms that simulate the human vascular system in an in vitro setup has been well studied and reported in the author’s previous publications.

a) Nomoni, Michelle, James M. May, and Panayiotis A. Kyriacou. "Novel polydimethylsiloxane (PDMS) pulsatile vascular tissue phantoms for the in-vitro investigation of light tissue interaction in photoplethysmography." Sensors 20.15 (2020): 4246.

b) Ferizoli, Redjan, et al. "Arterial stiffness assessment using PPG feature extraction and significance testing in an in vitro cardiovascular system." Scientific Reports 14.1 (2024): 2024.

After going through the current manuscript, I wonder what is the significance or advancement of this article in comparison with previous work.

3. Line 43: “While these procedures are accessible within clinical environments, they can induce patient anxiety” Please provide supporting references.

4. Figure 1: This figure has been reported in the paper with the title “A Bilateral In Vitro Model for Cardiovascular Disease Investigations Using Photoplethysmography Sensors” and thus can’t be reused here. I suggest redrawing a schematic diagram to avoid any ethical issue.

5. Figure 2: Please provide the cross-sectional image of a stained vessel that is sliced from both ends. In addition, a scale bar should be added to Figure 2.

6. Line 138: “This involved two grips holding the vessel in place and applying a controlled amount of tension by vertically drawing the sample apart until the material reached its breaking point.” Please provide sufficient technical details to allow others to reproduce the work. What is the length of the vessel between the two grips? How to define “a controlled amount of tension”? Also, what is the draw speed in this test?

7. Table 6: It is unclear why a reduction in tissue hardness was observed with increasing amounts of deadener.

Author Response

The authors express gratitude to the editor and reviewerS for their invaluable suggestions and comments, which have greatly helped improve the manuscript. We have addressed and incorporated their feedback into the revised manuscript accordingly. Below, you will find our responses addressing the reviewer’s comments.

  1. The abstract should state briefly the purpose of the research, the principal results and major conclusions. It should contain significant and quantitative findings.

Many thanks for your feedback. The authors have amended the abstract to reflect this.

  1. It seems that the methodology of fabricating custom silicone vessels and tissue phantoms that simulate the human vascular system in an in vitro setup has been well studied and reported in the author’s previous publications.
  2. a) Nomoni, Michelle, James M. May, and Panayiotis A. Kyriacou. "Novel polydimethylsiloxane (PDMS) pulsatile vascular tissue phantoms for the in-vitro investigation of light tissue interaction in photoplethysmography." Sensors 20.15 (2020): 4246.
  3. b) Ferizoli, Redjan, et al. "Arterial stiffness assessment using PPG feature extraction and significance testing in an in vitro cardiovascular system." Scientific Reports 14.1 (2024): 2024.

After going through the current manuscript, I wonder what is the significance or advancement of this article in comparison with previous work.

We would like to thank the reviewer for highlighting this valid point. We acknowledge the previous work, however, the method has advanced over the years. While the previous work was successful, those were initial results. Compared to Nomoni et al., the silicone gel used in this study (PlatSil silicone) offers adjustment of the silicone formulations to a finer degree, allowing for customisable custom vessels. Furthermore, the alternative silicone presented in the study has a built in scatterer, making it more comparable to the human vasculature. Compared to Ferizoli et al., this paper outlines mechanical tests (thickness, hardness, and elasticity) to report vascular properties for a full range of elastomer adjustments. The introduction section of the manuscript has been amended to explain and clarify this.

  1. Line 43: “While these procedures are accessible within clinical environments, they can induce patient anxiety” Please provide supporting references.

Thank you for the suggestion. Although this was a general statement, we could not identify a single manuscript to support it. Therefore, it has been removed, and the sentence has been restructured.

  1. Figure 1: This figure has been reported in the paper with the title “A Bilateral In Vitro Model for Cardiovascular Disease Investigations Using Photoplethysmography Sensors” and thus can’t be reused here. I suggest redrawing a schematic diagram to avoid any ethical issue.

The authors would like to thank the reviewer for this suggestion. Figure 1 has been revised, featuring a schematic diagram.

  1. Figure 2: Please provide the cross-sectional image of a stained vessel that is sliced from both ends. In addition, a scale bar should be added to Figure 2.

We would like to thank the reviewer for raising this valid point. It has been addressed by including cross-sectional images from both ends and integrating the scale bar into the figure. The caption of the figure has been modified to illustrate this.

  1. Line 138: “This involved two grips holding the vessel in place and applying a controlled amount of tension by vertically drawing the sample apart until the material reached its breaking point.” Please provide sufficient technical details to allow others to reproduce the work. What is the length of the vessel between the two grips? How to define “a controlled amount of tension”? Also, what is the draw speed in this test?

Many thanks for this valuable feedback. The authors have made revisions to the section, including specifying the length of the vessels between the grip and the draw speeds of the tests. This section has been reworded to enhance clarity.

  1. Table 6: It is unclear why a reduction in tissue hardness was observed with increasing amounts of deadener.

Thank you for bringing this to our attention, we have added more detail in the table captions to make it clear that this is the expected effect of deadener.

Reviewer 2 Report

Comments and Suggestions for Authors

This study describes the fabrication of custom made silicon vessels with varying mechanical properties. The vessels were embedded within custom made tissue phantoms and were evaluated in an in vitro setting utilizing photoplethysmography (PPG) reflectance sensors. Mechanical tests were conducted to analyse how different elastomer formulations affected vessel elasticity, wall thickness, diameter, and hardness. The primary objective was to explore how changes in silicone formulations influenced vessel properties and their correlation with features extracted from PPG signals from vessels with varying mechanical properties. It was observed that altering vessel elasticity significantly impacted PPG signal morphology, particularly reducing amplitude with increasing vessel stiffness (p < 0.001). This research lays the foundation for future studies to replicate healthy and unhealthy vascular systems.

Before being published, the following questions need to be addressed.

1. Some professional terms appear as full names only when they first appear, and then abbreviate them.

2. We think that making sure the font and size of the text on each figure are consistent will make it more beautiful. In particular, the font in Figure 3 is too small and it is low definition.

3. Why did the vertical axis of Figure 5, Figure 6, Figure 7, Figure 11 disappear? We think the authors should add the vertical axis to make the figures more intuitive.

4. What does the abscissa such as "0.6D and 0H" in Figure 7 mean?

5. What does "Finger Range" mean?

6. What is the unit of hardness in Table 5-Table 7? The authors should annotate in Figure 5.

Comments on the Quality of English Language

Moderate editing of English language required

Author Response

The authors express gratitude to the editor and reviewerS for their invaluable suggestions and comments, which have greatly helped improve the manuscript. We have addressed and incorporated their feedback into the revised manuscript accordingly. Below, you will find our responses addressing the reviewer’s comments.

  1. Some professional terms appear as full names only when they first appear, and then abbreviate them.

We appreciate your flagging this. We have thoroughly reviewed the manuscript and made the necessary corrections to the professional terms and abbreviations.

  1. We think that making sure the font and size of the text on each figure are consistent will make it more beautiful. In particular, the font in Figure 3 is too small and it is low definition.

Thank you for your feedback. We have reviewed all the Figures and can confirm that they are now consistent.

  1. Why did the vertical axis of Figure 5, Figure 6, Figure 7, Figure 11 disappear? We think the authors should add the vertical axis to make the figures more intuitive.

Thank you very much for your comment. We have amended the figures and added in the vertical axis.

  1. What does the abscissa such as "0.6D and 0H" in Figure 7 mean?

Thank you for bringing this to our attention. This ratio represents the additive relative to the total weight of equal parts of Part A and Part B, concerning both the deadener and hardener. We have addressed this in the figure caption and included it in the figure legend to enhance clarity. 

  1. What does "Finger Range" mean?

The authors would like to thank the reviewer for this comment. The finger range was referring to tissue found in the human finger. The legend of Figure 8 and caption have been amended to clarify this.

  1. What is the unit of hardness in Table 5-Table 7? The authors should annotate in Figure 5.

Thank you very much for bringing this to our attention. The hardness was measured using an OO Shore Durometer. The tables have been amended to make this clearer.

Reviewer 3 Report

Comments and Suggestions for Authors

In this manuscript the fabrication of customisable silicone vessels is described. The paper is well formatted and experimental section is well detailed. The following issues should be addressed:

-the introduction section should be improved with a deeper analysis of the literature on the fabrication of artificial vessels;

-the analysis of the fluid dynamics was carried out using a deionised-water based fluid. The properties of this fluid are not comparable with the blood (Casson fluid) since it should have a non-Newtonian behaviour. The evaluated performances could thus be alternated. I would suggest to repeat the analysis using a different fluid.

Author Response

The authors express gratitude to the editor and reviewerS for their invaluable suggestions and comments, which have greatly helped improve the manuscript. We have addressed and incorporated their feedback into the revised manuscript accordingly. Below, you will find our responses addressing the reviewer’s comments.

In this manuscript the fabrication of customisable silicone vessels is described. The paper is well formatted and experimental section is well detailed. The following issues should be addressed:

-the introduction section should be improved with a deeper analysis of the literature on the fabrication of artificial vessels;

Thank you for this feedback. We have included a deeper analysis of the literature on the fabrication of vessels, and mentioned disadvantages of the alternative techniques, in the introduction section.

-the analysis of the fluid dynamics was carried out using a deionised-water based fluid. The properties of this fluid are not comparable with the blood (Casson fluid) since it should have a non-Newtonian behaviour. The evaluated performances could thus be alternated. I would suggest to repeat the analysis using a different fluid.

Thank you for this valid comment, which brings an interesting discussion. The focus of the investigation was on manufacturing custom silicone vessels with varying stiffness properties and performing mechanical tests. We did also perform PPG signal validation and analysis using an in vitro setup containing a fluid mimicking the absorption spectra of blood. We agree that an in vitro investigation utilising non-Newtonian fluids would be beneficial to simulate the behaviour of human blood more accurately and have now mentioned this in the discussion. The fluid currently used helps to produce PPG signals in vitro without the difficulties of working with blood. As we advance going forward, we intend to integrate fluid with non-Newtonian behaviour into our future studies.

Round 2

Reviewer 2 Report

Comments and Suggestions for Authors

The authors have addressed my issues.

Reviewer 3 Report

Comments and Suggestions for Authors

Dear authors,

all the revisions have been addressed. I have no further comments.